# Efficacy of an Integrated Mobile Application System for Patients with Radiation Therapy: A Pilot Study

**DOI:** 10.3390/healthcare10091696

**Published:** 2022-09-05

**Authors:** Jaehyeon Park, Ji Woon Yea, Se An Oh, Jae Won Park

**Affiliations:** Department of Radiation Oncology, Yeungnam University College of Medicine, 170, Hyeonchung-ro, Nam-gu, Daegu 42415, Korea

**Keywords:** radiotherapy, mobile application, mobile healthcare

## Abstract

The use of IT applications for patients undergoing radiotherapy is limited. This study aimed to develop an integrated system for communication between patients and radiation oncologists using IT technology and report the first test results for the system “Assisted Radiation Oncology Mobile Application” (AROMA). This system consisted of a manager program, a server running on a PC, and a mobile application on a smartphone. A prospective survey was conducted to evaluate the usefulness of this system from October 2020 to January 2021. The survey consisted of a specific questionnaire on basic information and application use by the patients. The management program was designed such that the user (doctor) edits the treatment schedule, member (patient and doctor) information, self-management, disease information, and side effect questionnaire. The mobile application for patients consisted of the current schedule, treatment schedule calendar, side effect questionnaire, side effect management method, and disease information entered by the doctor. A total of 41 patients were enrolled in this study. The mean adverse event response time was 4.4 days. In the survey, the mobile application received positive views (8.6/10 points). Most responses related to the side effect reporting function (94%) and communication using the application (91%) were positive. Satisfaction with the application design and each menu item was high, with an average of ≥8 and ≥8.5 points in most cases, respectively. The survey showed good satisfaction with the design, operability, and reporting system. Therefore, the system can facilitate communication between patients and radiation oncologists in the future.

## 1. Introduction

Cancer is one of the main causes of death worldwide, with over 19.3 million newly diagnosed cases [1]. Cancer is the leading cause of death in Korea [2] and continually increasing deaths in the United States [3]. Radiation therapy is one of the three major cancer treatment methods along with surgery and chemotherapy. In Korea, the total estimated number of patients who received radiotherapy was 225,343 in 2013 [4]. Conventional fractionation radiation treatment usually takes up to 6–7 weeks, and self-management during this treatment period is very important. During radiation therapy, various side effects appear depending on the treatment site such as skin toxicity in breast cancer [5]. The radiation oncologist must carefully monitor these side effects; however, it may be inefficient to examine the patient every day using routine protocols. In places where the ratio of radiation treatment patients to doctors is high, such as in South Korea, patients may not be able to appropriately complain about side effects, and doctors may also have difficulty recording them due to short meeting times, once a week. The authors believe that this problem could be solved using information technology (IT).

There has been a great interest in healthcare using IT [6,7,8]. Dennis et al. reported that survival was improved by early detection of relapse and early palliative care initiation using a web application [9]. A few applications are also available for patients who receive radiotherapy [10,11,12,13]. However, these applications have been developed for specific cancers, such as head and neck cancer, or to help calculate the radiation dose. Radiation treatment is performed in various areas, and the interests and approaches to side effects and how to manage patients differ depending on the radiation oncologist. Therefore, we attempted to develop an integrated system that allows radiation oncologists to directly edit patient self-management and adverse event reporting methods as desired and provide them to patients. This study aimed to report the results of the first integrated system design/development and the first test results for patients.

## 2. Materials and Methods

This integrated system has been named the Assisted Radiation Oncology Mobile Application (AROMA). The AROMA system was developed for two-way communication between patients and doctors (Figure 1). AROMA is planned to run on iOS for iPhones (version 13 or higher) or Android for other mobile phones (Lollipop or higher). The program developer, three radiation oncologists, and one medical physicist participated in the development process: conception, development of beta version, testing, and program release.

### 2.1. Design and Development

This system consists of a manager program and server running on a PC and a mobile application on a smartphone (Figure 2). The manager program is designed primarily for use by doctors, with the following functions:Register patient/physician information;Manage patient care schedule;Enter the treatment area;Enter self-management method *;Enter the side effect questionnaire during treatment *;Enter disease information *.

* Steps 4–6 should correspond to the treatment area entered in Step 3.

The patient may use the mobile application, where the treatment schedule provided by the doctor, self-management method, and disease information can be viewed, and a side effect questionnaire can be filled out.

The manager program was developed using Visual Studio 2019, and the server was Visual Studio code. Mobile applications were developed using Android Studio for Android and Xcode for iOS (Table 1).

### 2.2. Validation

For a patient’s use in the management program, a radiation oncologist selected six treatment areas (brain, head and neck, breast, chest, abdomen, and pelvis) and entered the disease information, side effect management, and side effect questionnaires. Although management of the patient treatment schedule was also manually possible, for the convenience of the study, the management of the patient treatment schedule was automatically performed in conjunction with the hospital online chart system (OCS).

A prospective survey was conducted to evaluate the usefulness of this system in the Department of Radiation Oncology, Yeungnam Medical Center, Deagu, South Korea, from October 2020 to January 2021. The survey consisted of a specific questionnaire about the basic information and use of applications by the patients (Appendix A). We planned to enroll 40 patients, and the inclusion criteria were as follows: (1) above 18 years old; (2) smartphone users (iOS 13 version or higher, Android Lollipop or higher); (3) KPS 70 or higher; and (4) treatment site: brain, head and neck, breast, chest, abdomen, pelvis. In contrast, (1) patients without a smartphone, (2) patients who are inexperienced in using smartphones to the extent that it is difficult to use them even if they are taught how to use them, (3) patients with a short radiation treatment period of less than three weeks, and (4) patients receiving treatment in sites other than those entered in the application were excluded from this study.

The study was conducted in accordance with the Declaration of Helsinki and was approved by the Institutional Review Board (IRB) of the Yeungnam University Medical Center (approval code: YUMC 2020-09-009; date of approval), and the patients provided written informed consent.

## 3. Results

### 3.1. Management System

If the program is run on a PC, it automatically updates, and a login window pops up (Figure 3). Click buttons are used for patient and schedule management. Members, attending physician information, side effects, and disease information are tabbed under them (Figure 4). The side effect questionnaire can either be set to five grades according to the NCI-CTCAE version by default, or as a free questionnaire, or a score questionnaire; if a user clicks on the side effect information tab, he/she can add a treatment area by clicking the right mouse button on the left sidebar and add side effect questionnaires and disease management methods according to the treatment area (Figure 5). The side effect questionnaire entered by the patient can be checked in the data tab.

### 3.2. Mobile Application

The overall structure of the mobile application is shown in Figure 6. Membership registration is possible only when the information registered in the manager program is matched. If it is linked with OCS, the user can sign up through personal information registered with the hospital. On the first screen after logging in, the patient’s current schedule, treatment progress, and expected end date of treatment are displayed on the home screen. The treatment schedule calendar, side effect questionnaire input, side effect management method, and disease information are organized in the lower tables. The side effect questionnaire, disease management method, and disease information entered by the doctor are displayed, and the patient can view the information by scrolling up and down using the swipe function. Therefore, patients can browse their treatment schedule, manage their disease, and freely report side effects to doctors through the side effect questionnaires.

### 3.3. Validation

A total of 41 patients were enrolled in this study to assess the usability of mobile applications. Three patients refused to participate, and two other patients were using unsupported mobile phone devices and were excluded from the study. At the end of the treatment, two patients did not respond to the survey. Therefore, 33 patients were included in the survey. The median age group was 50 years, with the majority (25 (75.8%)) of patients being female. The treatment sites were breast (24), head and neck (3), chest (3), and pelvis (3).

One patient consented to the study but did not actually use the application. The average side effect response number was 8.8 times during the treatment period. The mean adverse event response time was 4.4 days. A total of 32 people answered more than two times (91.4%) (Figure 7).

In the survey conducted after the application was used, the perceptions regarding the use of mobile application technology were mostly positive (8.6 points/10 points). Most of the responses to the side effect reporting function (94%) and communication using the application were positive (91%). The intuition of application use and ease of navigation were all positive, except for non-response. Satisfaction with the design of the application and with each menu was high, with averages of >8 points and 8.5 points in most cases, respectively (Figure 8).

## 4. Discussion

There is a growing interest in using smartphones in healthcare [8,13,14,15]. Healthcare development using IT has mainly focused on medical personnel. In the field of radiation therapy, applications have been developed to aid in treatment planning or to assist with one-sided questionnaires. Therefore, we developed the AROMA system, considering that a system of two-way communication between the patient and the doctor was necessary.

Most patients reported their side effects. At the time of initial explanation regarding the application, we instructed patients to report their side effects once a week, and thereafter, most patients reported their side effects more than twice, even though no special instructions were given. Since side effects and treatment sites can be freely edited by the attending physician, it can be a useful tool to measure various patient-reported outcomes during radiotherapy. Recent clinical studies have emphasized the importance of patient-reported outcomes [16]. Routine web-based monitoring for NSCLC patients improves clinical outcomes due to early relapse detection [17]. An automated home monitoring system for daily reporting of severity improved chemotherapy-related side effects [18]. However, these study results have the disadvantage of not being applicable to various other diseases because they require the input of specific side effects for specific diseases.

South Koreans have the highest smartphone ownership rate worldwide [19]. In 2017, 89.5% of Koreans had smartphones, and 90.3% used the Internet. In fact, 99.4% of people use the Internet, of which 94.1% use a smartphone to access the Internet [20]. In South Korea, consultation time is as short as 4.5 min, thereby making it difficult to obtain a sufficient explanation from a doctor [21]. Cancer patients use the Internet to obtain information, and they tend to trust tips and information shared among patients [22,23]. Because Korean Internet portals (Naver and Daum) display cafes or blog posts as results when searching for information, it is easy for patients to acquire such information [24]. However, this information has not been scientifically verified and is often interspersed with promotional information or advertisements; therefore, patients are more likely to receive erroneous information [25]. Therefore, it is important to develop a system that delivers accurate medical information to patients and receives feedback. Therefore, this application was designed to enable the doctor to edit the disease information and management method during treatment directly in the manager program and provide them to the patient.

The patients’ preference for the AROMA application was generally high. Satisfaction with the design and usage environment was good. In particular, patients were highly satisfied with their ability to check their treatment schedules. Scheduling and treatment calendars may be useful in encouraging patients to use the application. In addition, the application can help adhere to appointment timings and pre-treatment preparations (fasting for abdominal treatment, bladder filling for pelvic treatment, etc.). This application allows doctors to communicate with patients regarding pre-treatment preparations and adherence during radiation therapy.

A retrospective study may be disadvantageous in terms of quality, which often depends on the accuracy of the data stored in medical records [26]. In particular, data on side effects are often missing from medical records [27]. The short consultation time in Korea, as mentioned earlier, increases the probability of generating incomplete medical records. This system enables the natural accumulation of data pertaining to side effects by allowing physicians to directly edit side effect questionnaires and present them to patients. In Korea, it is possible to obtain real-time reports of side effects that were not discovered at a regular consultation during the treatment period. These data may be of high value as they are prospectively recorded patient-reported outcomes.

This study had several limitations. First, the number of patients was small. Second, the relatively younger breast cancer patients accounted for the majority of the cohort because younger people tend to be somewhat familiar with application use and thus were included in the study. However, since this is a pilot study evaluating the development of a new system and its usefulness, we plan to assess the efficacy of its implementation with more patients in a future study. Furthermore, we plan to conduct a follow-up study on the satisfaction with using the integrated system among healthcare workers.

## 5. Conclusions

In conclusion, this system can be useful for management during radiation therapy, with high patient satisfaction. Patient–doctor communication in healthcare using mobile applications will have greater implications in the future. Based on this system, a distribution system is being developed, and further research will be needed to confirm the satisfaction of doctors as well as patients by applying it to various hospitals.

## Figures and Tables

**Figure 1 healthcare-10-01696-f001:**
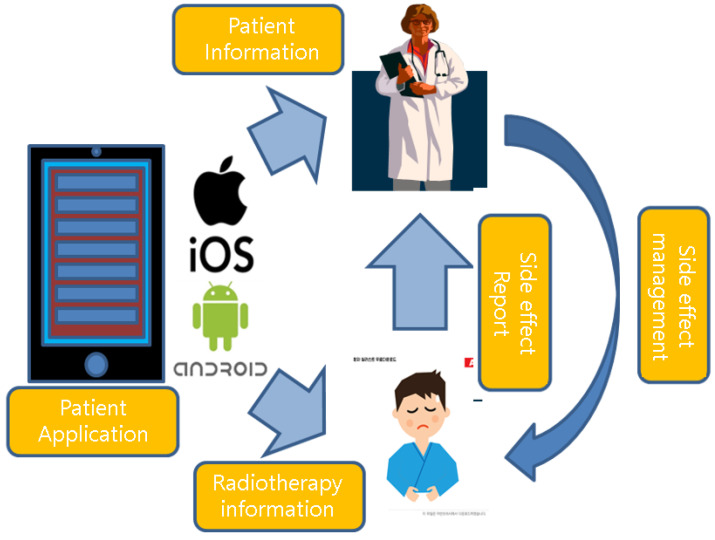
Two-way communication concept of the AROMA system.

**Figure 2 healthcare-10-01696-f002:**
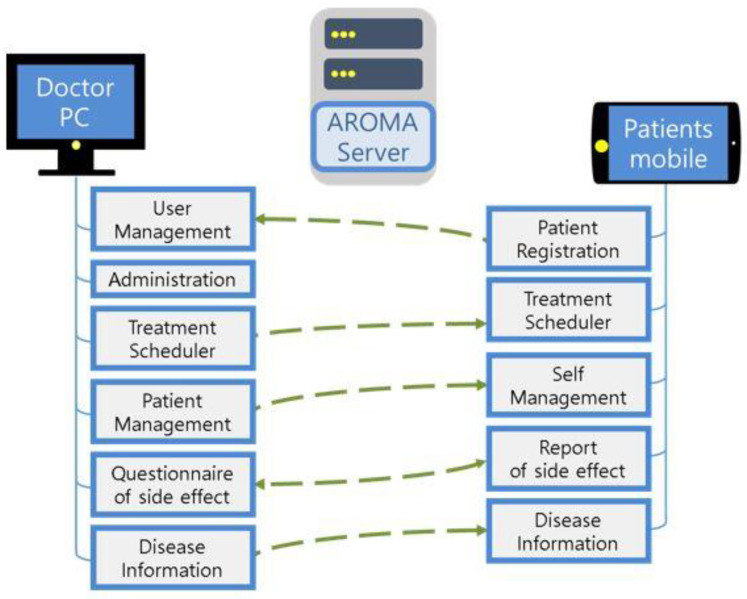
Basic concept of the AROMA system.

**Figure 3 healthcare-10-01696-f003:**
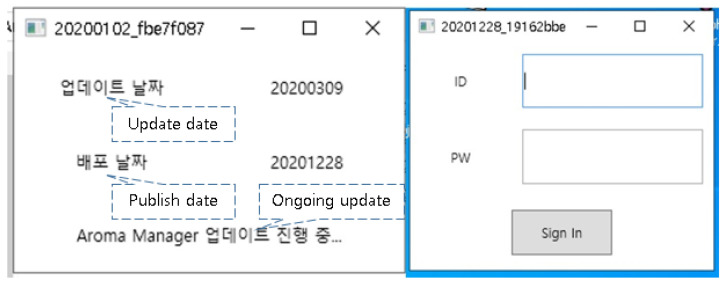
Log-in pop-up window of the manager program.

**Figure 4 healthcare-10-01696-f004:**
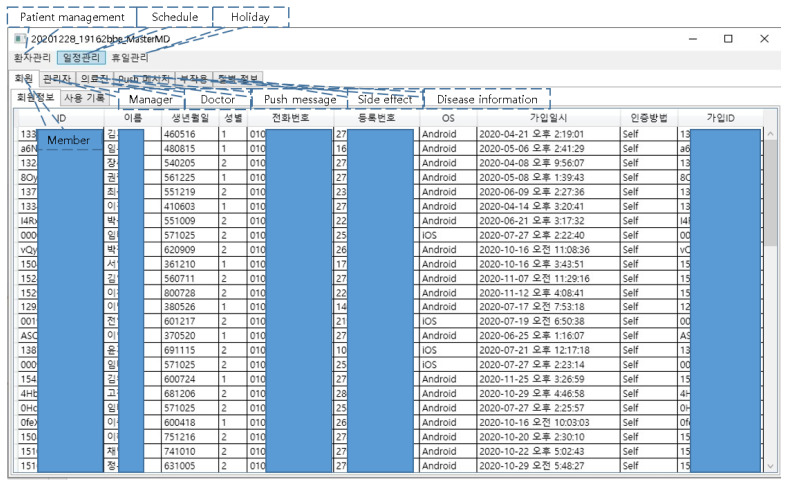
Start window screen of the manager program.

**Figure 5 healthcare-10-01696-f005:**
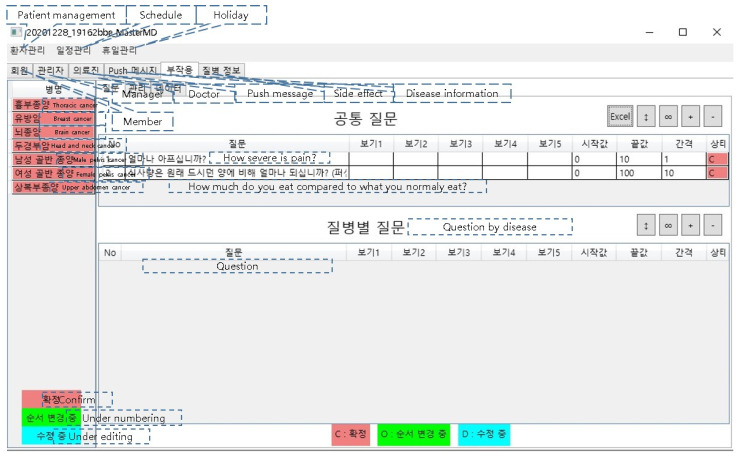
Side effect questionnaire window.

**Figure 6 healthcare-10-01696-f006:**
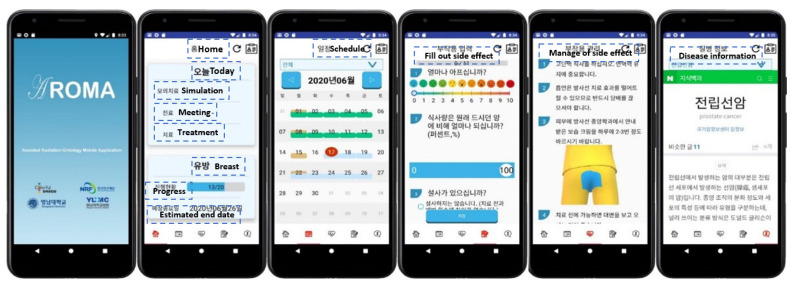
Overall structure of the mobile application.

**Figure 7 healthcare-10-01696-f007:**
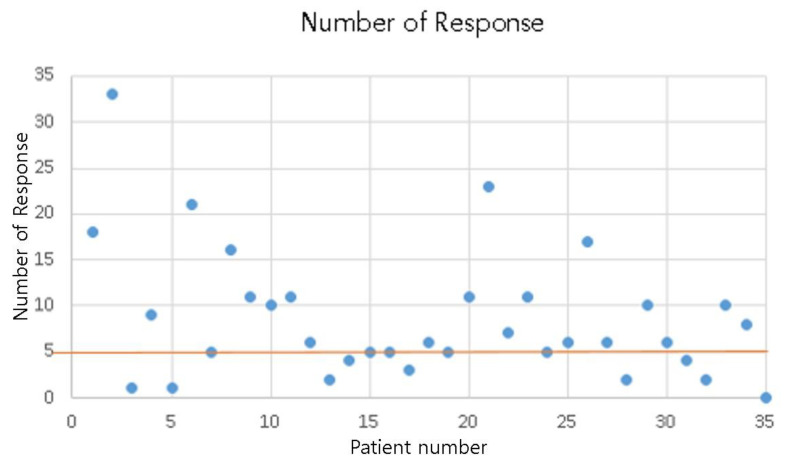
The number of side effect responses.

**Figure 8 healthcare-10-01696-f008:**
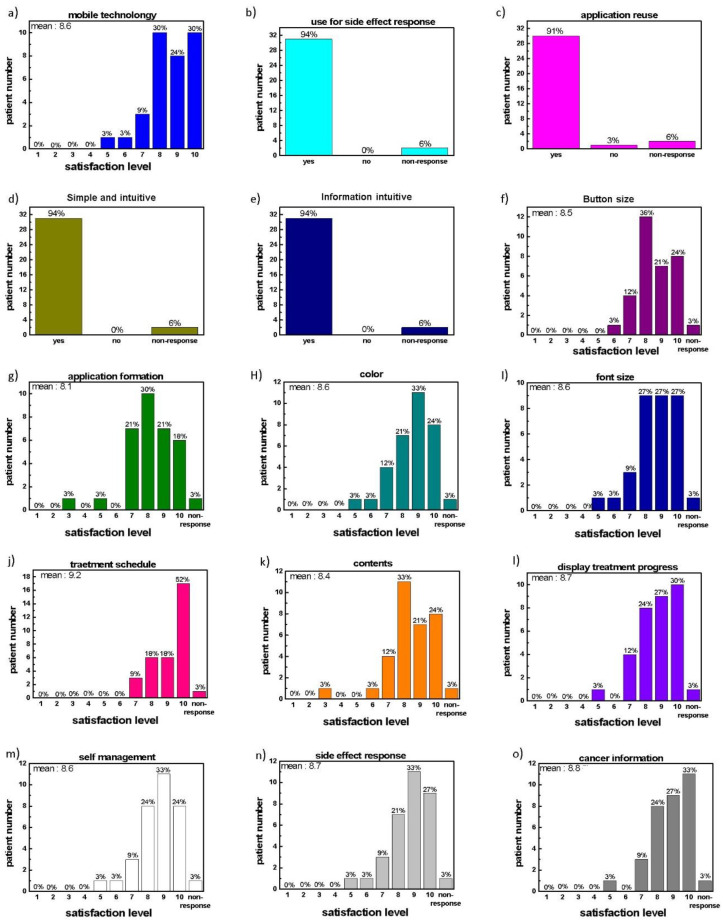
The items evaluated by questionnaire: (**a**) positive perception of mobile technology, (**b**) the possibility to report side effects, (**c**) the possibility of application re-use, (**d**) simple and intuitive application, (**e**) information intuitive, (**f**) button size, (**g**) formation of application, (**h**) color of application, (**i**) font size, (**j**) satisfaction with treatment schedule function, (**k**) satisfaction with contents, (**l**) satisfaction with treatment progression display, (**m**) satisfaction with self-management, (**n**) satisfaction with side effect reporting, (**o**) satisfaction with cancer information.

**Table 1 healthcare-10-01696-t001:** Development environment and minimal requirement.

	Client			Server
	Mobile Application (User)		Manager (Doctor)	
Name	AROMA M		AROMA	AROMA S
Minimal requirement	Android Lollipop	iOS 13	Windows 10 Pro	Windows 10 Pro
Language	Kotlin 1.4.32	Swift 5	C# 7.3	Go 1.15.5
Software	Android Studio 4.1.3`	XCode 12.4	Visual Studio 2019	Visual Studio Code 1.55.2

## Data Availability

Data are available upon request from the authors.

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
