# Peer review of "Efficacy of an Integrated Mobile Application System for Patients with Radiation Therapy: A Pilot Study"

_healthcare, 2022, doi:10.3390/healthcare10091696_

Round 1

Reviewer 1 Report

Thank you for including me in the review. The authors are investigating patient satisfaction with an app. The app is very simple and is a form of mobile questionnaires. Even though I don't doubt the usefulness of such an app, comparable apps will already exist and there is nothing really exciting about what this app can do. The evaluation is also meaningless.  I can't rule out for sure that the paper is meant to market an app. Possible conflicts of interest need to be presented more transparently. Even if this is really the smallest point of criticism: Chinese images could be translated in an English text, if necessary, in order to do justice to the international readership.  Overall, I do not find the paper interesting and would not recommend publication.

Author Response

Dear editor-in-chief and reviews,

I appreciate your attention and helpful comments, which have contributed significantly to the refinement of this manuscript. I would like to submit the revised manuscript (healthcare-1858605) “Efficacy of An Integrated Mobile Application System for Patients with Radiation Therapy: A Pilot Study” to be considered for publication in “Healthcare.”

Please find our point-by-point responses to the reviewer’s comments below.

Reviewer #1: Dear authors,

Thank you for including me in the review. The authors are investigating patient satisfaction with an app. The app is very simple and is a form of mobile questionnaires. Even though I don't doubt the usefulness of such an app, comparable apps will already exist and there is nothing really exciting about what this app can do.

à Thank you for your opinion. However, to our knowledge, at the time of conducting our research, there was no radiation oncology application that allowed doctors to freely edit and provide information directly to patients, and patients to communicate their symptoms to the doctor in real time.

The evaluation is also meaningless.  I can't rule out for sure that the paper is meant to market an app. Possible conflicts of interest need to be presented more transparently.

à Thank you for your opinion.

Please be assured that the research has not been conducted for any economic gain, and it is a project that was carried out with support from the National Research Foundation of Korea.

There is no conflict of interest here, and it is difficult to intervene with any subjectivity since all questionnaires were written by the patient themselves.

This application can be operated only by registering on the server directly at the hospital and having the patient's personal information, so it is not something that can be used by anyone in general. Therefore, our application is purely at the research stage, and not something that can be promoted to the market yet.

Even if this is really the smallest point of criticism: Chinese images could be translated in an English text, if necessary, in order to do justice to the international readership.

à Thank you for your recommendation.

However, the text in the figures is in Korean, not Chinese. We have added an English translation to the figure.

 Overall, I do not find the paper interesting and would not recommend publication.

à We apologize for not being able to generate interest for publication.

However, I humbly request you to consider our efforts to develop two-way communication between patients and doctors in the field of radiation oncology.

Reviewer 2 Report

This paper reported the evaluation of a newly developed online system for cancer patients to communicate with their doctors. This is an important first step towards improving medical care with modern technologies, and it is encouraging that the system is well-received among the patients.

The drawback in the study is the relative small number of patients in the survey and the overall subjective evaluation of the system. It is important to know whether the treatment is improved for the patients using other objective criteria, such as average time to relapse, average time spent with nurse/other healthcare personnel and number of complaints received, before and after implementation of the new system. The overall conclusion would be better appreciated with a larger cohort of users.

Author Response

Dear editor-in-chief and reviewers,

I appreciate your attention and helpful comments, which have contributed significantly to the refinement of this manuscript. I would like to submit the revised manuscript (healthcare-1858605) “Efficacy of An Integrated Mobile Application System for Patients with Radiation Therapy: A Pilot Study” to be considered for publication in “Healthcare.”

Please find our point-by-point responses to the reviewer’s comments below.

Reviewer #2: Dear authors,

This paper reported the evaluation of a newly developed online system for cancer patients to communicate with their doctors. This is an important first step towards improving medical care with modern technologies, and it is encouraging that the system is well-received among the patients.

à We greatly appreciate your positive comment.

The drawback in the study is the relatively small number of patients included in the survey and the overall subjective evaluation of the system. It is important to know whether the treatment improved for patients using other objective criteria, such as average time to relapse, average time spent with nurse/other healthcare personnel, and number of complaints received, before and after implementation of the new system. The overall conclusion would be better appreciated with the inclusion of a larger cohort of users.

à Thank you for your great opinion.

This is a pilot study on the production and use of an integrated mobile application system. As the reviewer pointed out, we are planning a prospective study, to build on our current results, with more patients. In addition, we are planning a study to establish a system that can be used in other hospitals to survey the satisfaction of healthcare workers.

However, we are aware of the limitations that were not mentioned in the manuscript, and the following text has been added to the manuscript:

à This study had several limitations. First, the number of patients was small. Second, the relatively younger breast cancer patients accounted for the majority of the cohort because younger people tend to be somewhat familiar with application use and thus were included in the study. However, since this is a pilot study evaluating the development of a new system and its usefulness, we plan to assess the efficacy of its implementation with more patients in a future study. Furthermore, we plan to conduct a follow-up study on the satisfaction of using the integrated system among healthcare workers.

Reviewer 3 Report

The authors present an interesting article concerning development of android-based application for patient-doctor communication during conventionally fractionated RT. The article is decently written, however, there are few minor issues to be addressed:

29-30 Cancer is among leading causes of death, but not the main cause of death worldwide (1st in only approx. 30% of the countries). It is the main cause of premature death in South Korea though, so that part of the sentence is correct.

31-32 Despite the fact that I’m a radiation oncologist, I find that claim a bit over the top. Radiation therapy is among the three main cancer treatment methods, along with surgery and systemic treatment.

33 ‘usually up to’, and keep in mind that we are consistently moving towards hypofractionation in many common cancers (i.e. breast, prostate)

36 It would be high inefficient in conv. Irradiation. I’m assuming it’s 1x/week for majority of the countries.

93 I presume that you did not exclude such patients, but did not include them to start off with. In this case – these are ‘exclusion criteria’. (the figure of speech which you used suggests post-hoc exclusion).

Figure 7 – I think that you were thinking of ‘no response’

Please elaborate on whether this is patented or freeware application, and how the reads could possibly use it.

 Besides, very interesting idea and nice study! Congratulations to the authors.

Author Response

Dear editor-in-chief and reviewers,

I appreciate your attention and helpful comments, which have contributed significantly to the refinement of this manuscript. I would like to submit the revised manuscript (healthcare-1858605) “Efficacy of An Integrated Mobile Application System for Patients with Radiation Therapy: A Pilot Study” to be considered for publication in “Healthcare.”

Please find our point-by-point responses to the reviewer’s comments below.

Reviewer #3: Dear authors,

The authors present an interesting article concerning development of android-based application for patient-doctor communication during conventionally fractionated RT. The article is decently written, however, there are few minor issues to be addressed:

à We greatly appreciate your positive comment.

29-30 Cancer is among leading causes of death, but not the main cause of death worldwide (1st in only approx. 30% of the countries). It is the main cause of premature death in South Korea though, so that part of the sentence is correct.

à Thank you for your recommendation. We have corrected the manuscript as follows:

“Cancer is one of the main causes of death worldwide, with over 19.3 million newly diagnosed cases”

31-32 Despite the fact that I’m a radiation oncologist, I find that claim a bit over the top. Radiation therapy is among the three main cancer treatment methods, along with surgery and systemic treatment.

à Thank you for your recommendation. We have corrected manuscript as mentioned by you

“Radiation therapy is one of the three major cancer treatment methods, along with surgery and chemotherapy”

33 ‘usually up to’, and keep in mind that we are consistently moving towards hypofractionation in many common cancers (i.e. breast, prostate)

à Thank you for your recommendation. We have corrected the manuscript, as suggested:

“Conventional fractionation radiation treatment usually takes up to 6-7 weeks, and self-management during this treatment period is very important”

36 It would be high inefficient in conv. Irradiation. I’m assuming it’s 1x/week for majority of the countries.

à Thank you for your recommendation. We agree with your opinion. Even in Korea, meeting times are usually once a week. Therefore, we have corrected the manuscript as follows:

“In places where the ratio of radiation treatment patients to doctors is high, such as in South Korea, patients may not be able to appropriately complain about side effects, and doctors may also face difficulty recording them due to short meeting times, once a week”

93 I presume that you did not exclude such patients, but did not include them to start off with. In this case – these are ‘exclusion criteria’. (the figure of speech which you used suggests post-hoc exclusion).

à Thank you for your opinion.

However, we enrolled patients and conducted the study according to the guidelines that we initially used. We did not intentionally exclude certain specific patients during the study period

Figure 7 – I think that you were thinking of ‘no response’

à Thank you for your opinion. However, Figure 7 shows the results of the questionnaire, and there were few non-responses.

Please elaborate on whether this is patented or freeware application, and how the reads could possibly use it.

à You can download this application and system, www.aroma.re.kr, freely.

Besides, very interesting idea and nice study! Congratulations to the authors

à We really appreciate your positive comment. Thank you once again.
